# Unassisted photoelectrochemical water splitting exceeding 7% solar-to-hydrogen conversion efficiency using photon recycling

Xinjian Shi[1,2,*], Hokyeong Jeong[3,*], Seung Jae Oh[3], Ming Ma[4], Kan Zhang[1], Jeong Kwon[4], In Taek Choi[5], Il Yong Choi[3], Hwan Kyu Kim[5], Jong Kyu Kim[3] & Jong Hyeok Park[1]

Various tandem cell configurations have been reported for highly efficient and spontaneous hydrogen production from photoelectrochemical solar water splitting. However, there is a contradiction between two main requirements of a front photoelectrode in a tandem cell configuration, namely, high transparency and high photocurrent density. Here we demonstrate a simple yet highly effective method to overcome this contradiction by incorporating a hybrid conductive distributed Bragg reflector on the back side of the transparent conducting substrate for the front photoelectrochemical electrode, which functions as both an optical filter and a conductive counter-electrode of the rear dye-sensitized solar cell. The hybrid conductive distributed Bragg reflectors were designed to be transparent to the long-wavelength part of the incident solar spectrum ($\lambda > 500$ nm) for the rear solar cell, while reflecting the short-wavelength photons ($\lambda < 500$ nm) which can then be absorbed by the front photoelectrochemical electrode for enhanced photocurrent generation.

[1] Department of Chemical and Biomolecular Engineering, Yonsei University, 50 Yonsei-ro, Seodaemun-gu, Seoul 120-749, Republic of Korea. [2] Department of Mechanical Engineering, Stanford University, Stanford, California 94305, USA. [3] Department of Materials Science and Engineering, Pohang University of Science and Technology, 77 Cheongam-ro, Nam-gu, Pohang 790-784, Korea. [4] Department of Chemical Engineering, School of Chemical Engineering and SKKU Advanced Institute of Nano Technology (SAINT), Sungkyunkwan University, Chunchun-dong, Suwon 440-746, Korea. [5] Global GET-Future Lab, Department of Advanced Materials Chemistry, Korea University, 2511 Sejong-ro, Sejong 339-700, Korea. * These authors contributed equally to this work. Correspondence and requests for materials should be addressed to J.K.K. (email: kimjk@postech.ac.kr) or to J.H.P. (email: lutts@yonsei.ac.kr).

Conventional photoelectrochemical (PEC) tandem devices for unassisted solar water splitting systems consist of a front photoelectrode for water splitting and a rear photovoltaic cell[1,2]. Since the design of a tandem cell configuration for water splitting was first proposed[1–3], a variety of studies have been reported[4,5], and various photoanode[4,6]/photovoltaic[5,7–13] combinations have been tested to optimize the performance and efficiency. For example, Shi *et al.* reported a wireless monolithic photoanode/dye-sensitised solar cell (DSSC) tandem device[8] that could realize unassisted solar water splitting with a high solar-to-hydrogen (STH) efficiency (5.7%). Nevertheless, the efficiency must be further improved to greater than 10% to reach a viable level for commercialization. To achieve an STH efficiency above 10%, the photoelectrode of the front cell in tandem devices needs to be absorptive to effectively harvest photons from the sun while also being transparent enough to feed the rear cell for unassisted operation.

Typically, the short-wavelength component of solar radiation is mainly absorbed by the PEC photoelectrode, and the remaining long-wavelength component can be harvested by the rear solar cell (for example, DSSC). Increasing the transparency of the PEC photoelectrode for the rear cell results in a reduced photocurrent generation from the front PEC cell. This results in an unsatisfactory compromise with regard to the transparency to find an acceptable balance between the light absorption by the front and rear photoelectrodes via the use of a semi-transparent PEC photoelectrode, resulting in an unavoidable limited utilization of the solar spectrum by the rear cell[6–9]. Incorporating an optical filter called a distributed Bragg reflector (DBR)—consisting of multiple layers of alternating materials with high- and low-refractive indices that can reflect the short-wavelength component of solar radiation for photon recycling by the PEC photoelectrode while transmitting the long-wavelength component for the rear solar cell—into the tandem cell would be very beneficial for an effective utilization of sunlight. However, the conventional dielectric DBR framework has a non-conductive nature, which limits its applicability for monolithic wireless tandem systems in which it should act as a conductive substrate for counter-electrode of the rear solar cell as well. A number of studies on the combination of two thin-film materials that are conductive and transparent, yet have high refractive-index contrast have been performed[14–16], however, their application in PEC tandem cell with desired optical and electrical properties is still immature and needed to be further studied.

Here, we demonstrate unassisted PEC solar water splitting tandem devices with photon-recycling hybrid conductive DBRs (cDBRs). The hybrid cDBR structures consist of alternating dense and porous layers of conductive indium-tin-oxide (ITO) stacks on a conventional $TiO_2/SiO_2$ DBR framework. These hybrid cDBRs are optimized to exhibit a high reflectance for wavelengths shorter than 500 nm to recycle photons for the front PEC cell and a high transmittance above 500 nm for the effective operation of the rear DSSC while maintaining conductivity to serve as the counter-electrode of the real cell. The PEC/DSSC tandem device with the hybrid cDBRs shows unassisted hydrogen evolution with an STH efficiency of 7.1%, which is the best performance obtained to date from an n-type oxide-based PEC cell.

## Results

### Concept and design of hybrid cDBRs.
A schematic description of the PEC/DSSC tandem water splitting device with the bipolar electrodes is presented in Fig. 1. For the left part, one side is a $BiVO_4/WO_3$ photoanode on fluorine-doped tin oxide (FTO), and the other side is a hybrid cDBR with surface Pt coating (counter of DSSC). For the right part, one side is dye/$TiO_2$ (anode of

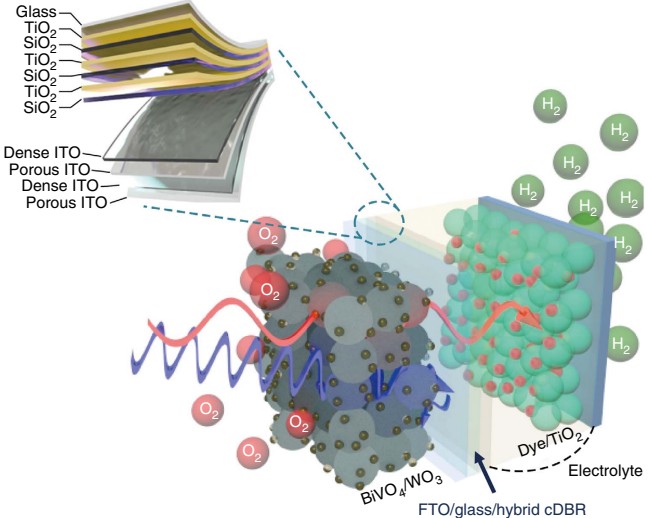

**Figure 1 | Schematic illustration of the PEC/DSSC tandem device.** For the left part, the back side of the photoanode is a hybrid cDBR consisting of an alternating dense and porous ITO layers on top of $TiO_2/SiO_2$ stack. The hybrid cDBR reflects short-wavelength light $< 500$ nm (represented by the blue arrow) while transmitting longer-wavelength light (represented by the red arrow). Pt island is coated on its top. For the right part, dye/$TiO_2$ photoanode and Pt layer are placed at the two opposite sides.

DSSC) and the other side is Pt layer as the counter of the tandem device. The hybrid cDBR consists of alternating layers of dense- and porous-ITO layers on the $TiO_2/SiO_2$ DBR stack with the stopband edge near 500 nm. The fabrication process is described in detail in the Methods section. When the sunlight is incident to the tandem device, the $BiVO_4/WO_3$ photoanode mainly absorbs short-wavelength photons, both from the incident light and the recycled light reflected by the hybrid cDBR. The longer wavelength component of the solar irradiation transmits through the $BiVO_4/WO_3$ photoanode as well as the hybrid cDBR and is absorbed by the dye/$TiO_2$ electrode in the rear DSSC. The optical properties of the hybrid cDBR structures were optimized using a genetic algorithm (GA)[17,18] and experimentally realized using the oblique-angle deposition (OAD) technique to tune the refractive index of the conductive ITO on demand[19]. The measured refractive indices of the dense- and porous-ITO films were 1.92 and 1.45 at $\lambda = 550$ nm, respectively, indicating sufficient refractive index contrast for the DBR function. To identify an optimum hybrid cDBR structure, the absorptance spectra of the $BiVO_4/WO_3$ photoanode and the dye/$TiO_2$ electrode used in the DSSC (Fig. 2a) were taken into account in a figure of merit for the GA optimization as shown in equation (1),

$$\text{FOM} = R_{<500 nm} + T_{>500 nm}$$

$$= \frac{\int_{\lambda=300nm}^{\lambda=500nm} w(\lambda)R(\lambda)}{\int_{\lambda=300nm}^{\lambda=500nm} w(\lambda)} + \frac{\int_{\lambda=500nm}^{\lambda=800nm} w(\lambda)[1 - R(\lambda)]}{\int_{\lambda=500nm}^{\lambda=800nm} w(\lambda)} \quad (1)$$

where $R(\lambda)$ is the reflectance of the hybrid cDBR and $w(\lambda)$ is a weighting factor based on the absorptance spectra of the $BiVO_4/WO_3$ photoanode and the dye/$TiO_2$ electrode in the DSSC. Note that instead of using ITO-only cDBRs, the hybrid structures consisting of ITO cDBRs stacked on the conventional three pairs of the $TiO_2/SiO_2$ DBR stack were designed to maximize the optical filtering property while maintaining the conductivity in the ITO cDBR. The GA-optimized single and double pairs of porous/dense ITO stacks (two-layer and four-layer ITO) on the $TiO_2/SiO_2$ DBR structure showed the optimal optical properties (Table 1); thus, these designs were chosen for the fabrication.

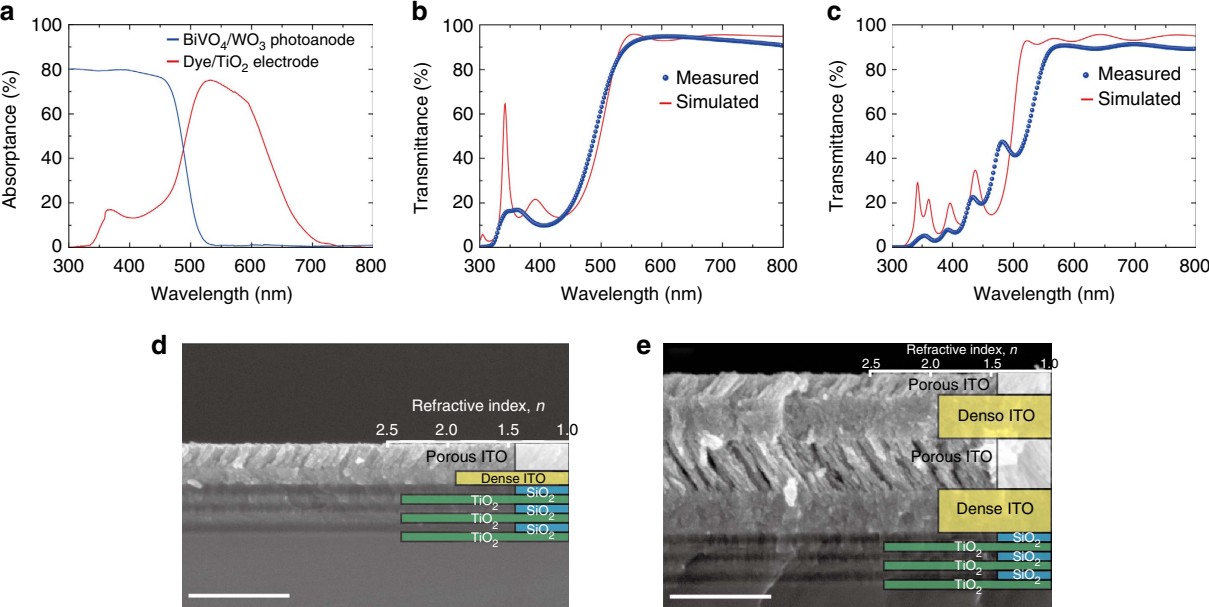

**Figure 2 | The hybrid cDBR.** (**a**) Absorptance spectra of the BiVO$_4$/WO$_3$ photoanode (blue solid line) and the dye/TiO$_2$ electrode measured behind the BiVO$_4$/WO$_3$ photoanode (red solid line). Measured (blue dotted lines) and simulated (red solid lines) transmittance spectra of the hybrid cDBR with (**b**) two- and (**c**) four-layer ITO. Cross-sectional SEM images of the hybrid cDBR with (**d**) two- and (**e**) four- layer ITO, showing clear interfaces between the layers. Scale bar, 500 nm.

## Table 1 | Optimized structures of the hybrid cDBRs with 2- and 4-layer ITO.

|  | cDBR with 2-layer ITO | cDBR with 4-layer ITO |
|---|---|---|
| Structure |  | Porous ITO, 117 nm |
|  |  | Dense ITO, 197 nm |
|  | Porous ITO, 127 nm | Porous ITO, 249 nm |
|  | Dense ITO, 94 nm | Dense ITO, 203 nm |
|  | SiO$_2$, 65 nm |  |
|  | TiO$_2$, 32 nm |  |
|  | SiO$_2$, 60 nm |  |
|  | TiO$_2$, 50 nm |  |
|  | SiO$_2$, 64 nm |  |
|  | TiO$_2$, 25 nm |  |
|  | Glass substrate |  |
| $R_{300-500nm}$ | 40.52% | 40.61% |
| $R_{500-800nm}$ | 8.67% | 8.09% |
| FOM | 1.3186 | 1.3238 |

cDBR, conductive distributed Bragg reflector; FOM, figure of merit; ITO, indium-tin-oxide. The corresponding average reflectance $R$ for the short-wavelength (300–500 nm) and long-wavelength (500–800 nm) regions.

Detailed GA optimization calculations and the experimental procedures for designing the hybrid cDBR stacks are provided in Supplementary Figs 1–3 and Supplementary Note 1.

**Fabrication and characterization of hybrid cDBRs.** The measured transmittance spectra (blue dotted lines) of the fabricated hybrid cDBR structures were compared with simulated values (red solid lines), as shown in Fig. 2b,c. The measured spectra were in good agreement with the simulated results, except for the region near 360 nm because of the absorption effect of the glass substrate. These results confirm that the fabricated hybrid cDBR structures with two- and four-layer ITO stacks predominantly reflect the short-wavelength component of the solar radiation to recycle high-energy photons for the PEC photoanode yet have a high transmittance in the long-wavelength component for

effective absorption by the dye/TiO$_2$ electrode in the rear DSSC, as indicated by the figure of merit in the GA optimization. Figure 2d,e showed cross-sectional scanning-electron microscopy (SEM) images of the hybrid cDBR with two- and four-layer ITO stacks, respectively, with the refractive index profiles of the individual layers measured by ellipsometry. These SEM images demonstrated that the fabricated structures had clear interfaces between each layer, whose thickness was well controlled as designed by the GA optimization. The fabricated hybrid cDBRs with two- and four-layer ITO stacks showed high conductivity, with sheet resistances of 20.4 and 35.0 $\Omega$ sq$^{-1}$, respectively, indicating a moderate electrical property as an electrode in addition to a superior optical property for the effective utilization of the solar radiation. Because both structures owned very similar optical results, the hybrid cDBR with two-layer ITO that showed a lower sheet resistance was chosen for the PEC tandem cell fabrication.

**PEC performances.** Bipolar conductive electrodes composed of conventional FTO and the hybrid cDBR were used in the tandem system[7,8], in which the photoanode materials mesoporous (W, Mo)-doped BiVO$_4$/WO$_3$ were placed on the FTO side and a thin Pt layer was coated on the hybrid cDBR stack. The typical cobalt electrolyte-based DSSC with a high open circuit potential ($V_{oc}$) was selected as the rear photovoltaic[20] to assemble the whole device. The PEC performances of photoanodes were measured under three-electrode system (Supplementary Fig. 4) and the performances of the tandem devices were evaluated using a two-electrode electrochemical configuration. For two-electrode measurement, before fabrication of the tandem device, the performances of the front PEC photoanode and the rear DSSC were evaluated independently. Notably, the J–V curves of the rear DSSC were obtained under the condition of 1 Sun illumination filtered by the semi-transparent front PEC photoanode with/without cDBR. The operating current density ($J_{op}$) and the operating potential ($U_{op}$) of the PEC tandem devices could be defined based on the individual two-electrode J–V curves (Fig. 3a)[21,22]. It can be seen that although the performance of

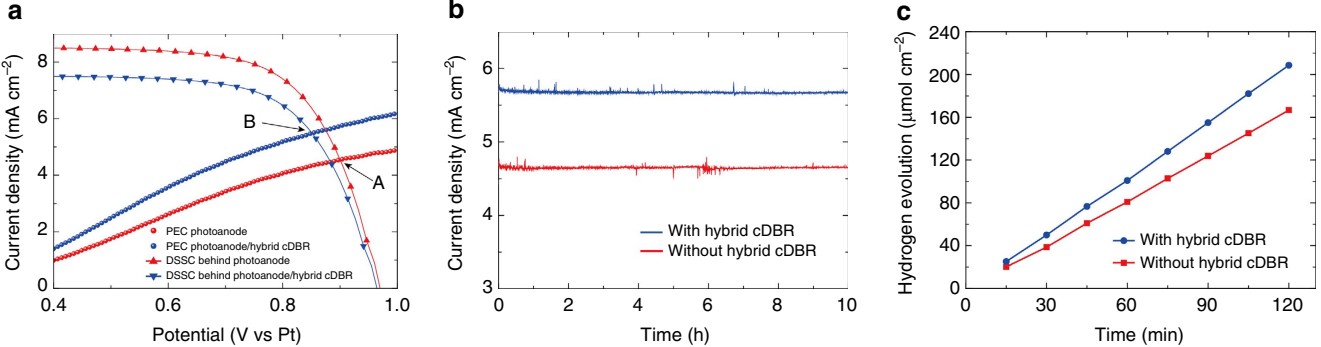

**Figure 3 | PEC performances. (a)** The measured PEC performance in a two-electrode system under AM1.5G illumination. The red dots correspond to the *J–V* curve of photoanode without the hybrid cDBR stack, and the red dotted line corresponds to that of DSSC (checked behind photoanode). The blue dots indicate the photoanode/hybrid cDBR data, and the blue dotted line indicates the DSSC data (determined behind the photoanode/hybrid cDBR). Points A and B define the operating current density ($J_{op}$) and the operating potential ($U_{op}$), respectively, for the tandem device without/with the hybrid cDBR interlayers; point B realizes a higher $J_{op}$ under a lower $U_{op}$. **(b)** The long-term current density versus time (*J–t*) curve of the tandem device with (blue) and without (red) the hybrid cDBR stack. **(c)** The hydrogen evolutions of the tandem device with (blue) and without (red) the hybrid cDBR stack.

the rear DSSC was slightly lower under the filtered 1 Sun illumination condition due to the front PEC photoanode with the hybrid cDBR, the performance of the front photoanode was enhanced with the help of the hybrid cDBR. As a result, the higher operating current density $J_{op}$ of the tandem system is achieved under a lower operating potential $U_{op}$, as illustrated by the points A and B. This result indicates that the restricted DSSC performance with the presence of the hybrid cDBR is a minor factor in the enhanced overall performance. Detailed information from theoretical studies about the effect of the hybrid cDBR on the light absorption of the front photoanode and rear DSSC, which further affects their respective performances, can be found in Supplementary Figs 5–7 and Supplementary Note 2.

The long-term current density versus time (*J–t*) performance of the PEC tandem devices with and without the hybrid cDBR were compared (Fig. 3b); the current density was well matched with the $J_{op}$ (Fig. 3a), with highly stable behaviours for 10 h for both the tandem devices with or without cDBR. We confirmed the corresponding evolution of hydrogen (Fig. 3c) from the well-mixed gases from both anode and cathode chambers. Besides, in order to characterize the gas crossovers for our membrane-free system, we also sampled and analysed the gases from two chambers, respectively[23], as shown in Supplementary Fig. 8. During the measurement the potential drop due to the electrolyte resistive loss could be quite small, at $10^{-2}$ V level based on our tandem device operating current density, as well as the near-neutral pH electrolyte used[24]. In addition, the use of wireless tandem device for gas evolution in this work reduced the distance between the working and counter, and decreased the solution loss in further[25,26]. According to the results in Fig. 3, a steady photocurrent density of 5.75 mA cm$^{-2}$ without any additional bias was achieved by the PEC tandem device with hybrid cDBR, which is 1.1 mA cm$^{-2}$ greater than the value for the device without hybrid cDBR. The unassisted STH efficiency of the PEC tandem device with hybrid cDBR was estimated to be ~7.1% according to the equation $\eta_{STH} = J_{op} \times 1.23/P$ ($P$ is the power of the illuminating light)[27,28].

## Discussion

In summary, we demonstrated the enhanced STH conversion efficiency of an unassisted PEC solar water-splitting tandem cell incorporated with a hybrid cDBR, which serves as both an optical filter to effectively utilize photons as well as a counter-electrode of the rear DSSC. Through photon recycling by the hybrid cDBR,

the issue of trade-off between the light absorption and the transmittance of the front PEC photoanodes could be solved. As a result, we demonstrated a 7.1% STH conversion efficiency without any external potential, which is the highest value reported to date for a PEC/solar cell tandem device. In this study, the optical properties of the hybrid cDBR were designed by GA optimization to be compatible with the photoanode (BiVO$_4$/WO$_3$) and the photovoltaics (DSSC). As a versatile technology exhibiting application-specific optical properties by GA optimization, the hybrid cDBR can be applied to many different PEC solar water-splitting systems and photovoltaics.

## Methods

**Preparation of WO$_3$ layer.** The porous and highly transparent WO$_3$ host layer was prepared by surfactant-assisted synthesis. The precursor solution of H$_2$WO$_4$ (10:1 PEG:W weight ratio)[7] was prepared by slowly dissolving 0.9 g of tungsten powder in 5 ml of H$_2$O$_2$ (35%, Junsei) and stirring the solution for 6 h. Then, 10 ml of 2-propanol ((CH$_3$)$_2$CHOH, Junsei) was added to the solution, and stirring was continued for one additional day. Finally, 8 ml of polyethylene glycol (PEG, Aldrich) was added to the solution as a surfactant to provide a mesoporous film morphology. The substrate, FTO (TEC-8, Pilkilton) glass, was cleaned in a mixture of ethanol and acetone (1:1 by vol%) for 10 min with sonication and then immersed in a mixed H$_2$SO$_4$/H$_2$O$_2$ solution for another 10 min for surface hydrophilization. The prepared WO$_3$ precursor solution was dropped onto the FTO substrate, which was placed on an optic table. The sample was then dried at room temperature for ~20 min to obtain a uniform distribution of the solution on the top of the FTO substrate. The samples were annealed using a temperate gradient from 300 to 550 °C, in which the temperature was increased at a rate of 10 °C min$^{-1}$ followed by annealing for 30 min at 550 °C. To obtain an optimal WO$_3$ layer, 20 µl of precursor solution was spread on the pre-defined FTO (1.5 × 1.5 cm$^2$) followed by the drying and annealing process for one cycle, and totally two cycles were done and used for the heterojunction fabrication.

**Fabrication of (W, Mo)-doped BiVO$_4$ layer on WO$_3$ layer.** The BiVO$_4$ precursor solution was prepared from a mixture of bismuth nitrate hexahydrate (BiN$_3$O$_9$·5H$_2$O, 99.99% Aldrich), vanadyl acetylacetonate (C$_{10}$H$_{14}$O$_5$V, 98% Aldrich) and sodium molybdate (Na$_2$MoO$_4$, Aldrich) at a mole ratio of 100:96:4 that was added to a solution of 1:0.12 acetylacetone (C$_5$H$_8$O$_2$, Fluka) and acetic acid (CH$_3$COOH, 99.70%, DAE). After sonication for 10 min, a dark green solution was obtained. Care was taken such that visible floating matter was not present, and the solution was used within 1 day after preparation to avoid sedimentation and prevent void filling of the mesoporous WO$_3$ bottom layer. The final mole concentration of Bi was 0.09 M. A total of 100 µl of the BiVO$_4$ precursor solution was dropped on the as-prepared mesoporous WO$_3$ bottom layer and kept for 1 min for solution penetration with the following spin coating process (5 s, 500 r.p.m. plus 30 s, 1,500 r.p.m.). The samples were then dried at 100 °C for 10 min, and then annealing at 300, 400, 500 °C for 5 min, respectively. During the annealing processes, W could be naturally doped into the BiVO$_4$ because of the intimate contact of the two components in the embedded structure. This process was conducted in a dry and well-ventilated environment and repeated five times. Finally, the samples were annealed in a box furnace at 500 °C for 2 h.

**FeOOH and NiOOH co-catalyst surface modification.** The FeOOH/NiOOH layer was deposited as follows. Using photo-assisted deposition (1 Sun, AM 1.5G), FeOOH was deposited in a 0.1 M $FeSO_4$ (iron(II) sulphate heptahydrate, 99%, Aldrich) solution under 0.25 V versus Ag/AgCl for 13 min. NiOOH was then deposited in 0.1 M $NiSO_4$ (nickel(II) sulphate hexahydrate, 99%, Aldrich, adjusted pH to 6.7 with basic solution) under 0.11 V versus Ag/AgCl for 6 min. Finally, one additional NiOOH layer was formed by electrodeposition in 0.1 M $NiSO_4$ under 1.2 V versus Ag/AgCl for 1.5 min.

**Optimization and fabrication of the hybrid cDBRs.** The structures of the hybrid cDBR were designed by two-step GA optimization. First, the dielectric DBR structure composed of a $TiO_2$/$SiO_2$ multilayer stack was optimized. Wavelength-dependent refractive index profiles of sputter-deposited $TiO_2$ ($n = 2.40$ at $\lambda = 550$ nm) and $SiO_2$ ($n = 1.45$ at $\lambda = 550$ nm) measured by ellipsometry were used. Then, ITO cDBR stacks were optimized on the top of the dielectric DBR after fixing the previously optimized structure. The measured refractive indices and the extinction coefficients of the dense and the porous ITO layers were used for the optimization of the cDBRs (Supplementary Fig. 1a).

The hybrid cDBR stack was fabricated on the back side of the photoanode. First, a $TiO_2$/$SiO_2$ multilayer stack was deposited on the back side of the FTO glass by radio-frequency magnetron sputtering with targeted thicknesses. On the top of this dielectric DBR stack, the ITO cDBR stack was fabricated by OAD using electron-beam evaporation. The dense ITO layer was formed without substrate tilting ($\theta_{OAD} = 0°$), and the low-refractive index porous ITO layer was formed by OAD with a substrate tilt angle of 70° ($\theta_{OAD} = 70°$). To improve the optical transparency and conductivity of the ITO-based cDBR stack, the hybrid cDBR samples were annealed in ambient oxygen at 550 °C for 1 min using a rapid thermal annealing system.

**Fabrication of dye-coated $TiO_2$ electrode of DSSC.** A 40 mM titanium tetra-chloride ($TiCl_4$, Aldrich) solution was prepared, and cleaned FTO substrates were immersed in this solution for 30 min (70 °C). Using the screen-print technique, nanocrystalline transparent $TiO_2$ electrodes ($TiO_2$ paste-Dyesol 18NR-T) were deposited onto the FTO glass substrates and then annealed at 500 °C for 30 min. The thickness of this layer was confirmed using an Alpha-step 250 surface profilometer (Tencor Instruments). The as-prepared $TiO_2$ electrodes were annealed at 500 °C for 30 min and then immersed in a THF/ethanol (v/v, 2:1) solution containing 0.3 mM JK-306 dye and 0.3 mM 4-[bis(9,9-dimethyl-9H-fluoren-2-yl)amino]benzoic acid coadsorbent for 12 h at room temperature.

**Preparation of counter-electrode of the tandem device.** After the fabrication of the DSSC photoanode on one side of FTO substrate, the other side of glass was coated with Pt layer by e-beam evaporation. This Pt layer is used as the hydrogen evolution reaction (HER) catalyst in the PEC tandem device (work as the counter of the whole tandem device). The dye-coated $TiO_2$ electrode side and the Pt side were electrically connected with silver paste.

**Assembly of the PEC tandem device.** After fabrication of hybrid cDBR stack at the back side of PEC photoanode, it was coated with an $H_2PtCl_6$ solution and heated at 300 °C for 30 min for its use as the counter-electrode of the DSSC. The two sides of glass (PEC photoanode on FTO and Pt islands on top of cDBR) were also electrically connected with silver paste. Two plates, the front PEC photoanode/Pt and the rear dye-adsorbed $TiO_2$/Pt (Pt for HER), were assembled using a 60-μm-thick Surlyn spacer (Surlyn-1702, Dupont). An electrolyte solution of 0.22 M $Co(bpy)_3(BCN_4)_2$, 0.05 M $Co(bpy)_3(BCN_4)_3$, 0.1 M $LiClO_4$ and 0.8 M 4-*tert*-butylpyridine in acetonitrile was then injected into the sandwiched cell through a hole drilled in the rear $TiO_2$ electrode. The hole was sealed with a Surlyn spacer and a cover glass. All of the exposed conductive glass or conductive material such as silver paste were totally covered by super glue with drying process before immersing into the electrolyte for measurement.

**Characterizations and additional measurements.** The evolution of hydrogen and oxygen was quantified with a gas chromatograph equipped with a thermal conductivity detector and a 5-Å molecular sieve column. A phosphate buffer solution (aqueous 0.1 M $Na_2HPO_4$, pH 9.2, and $NaH_2PO_4$, pH 4.0) after Ar-purging was used to adjust the pH of the electrolyte to 6.9. The membrane-free solar-driven water splitting system was used for gas evolutions. During the measurements, all cell compartments were completely sealed with rubber septum and glycerine, after then the gases generated from two chambers were inter-mittently sampled and analysed individually to calculate the gas crossovers. For more accurate value of the whole generated hydrogen, the evolved gases from two chambers were well mixed before analysed. The PEC performance of the photoanode was determined by using the potentiostat with different electrode system. For the two-electrode measurement, the Pt foil acted as both the reference and counter-electrode. For the three-electrode measurement, Ag/AgCl electrode was used as reference electrode and Pt foil was used as the counter-electrode. *J*–*V* curves were obtained under 1 Sun and AM 1.5 G using a solar simulator for irradiation (a 300-W xenon lamp (Newport)). The electrolyte was a 0.5 M $Na_2SO_4$

solution (adjusted with buffer solution). A silicon reference cell (Fraunhofer ISE, Certificate No. C-ISE269) was used to calibrate the exact 1 Sun condition. To measure the performances of the PEC tandem devices, the Pt (HER catalyst) on the back side of the rear electrode was removed. The working lead of the potentiostat was connected to the dye-coated $TiO_2$ electrode, and the counter and reference lines were connected to the separated platinum foil. The dimension of the used Pt foil was $1.5 \times 1.5$ cm$^2$. The optical properties during the experiments were measured using a ultraviolet–visible spectrophotometer (UV-2401 PC, Shimadzu). Morphology analyses of the samples were performed using field-emission SEM (JSM-7000F).

**Data availability.** Data supporting the findings of this study are available within the article and its supplementary information files and from the corresponding author upon reasonable request.

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

## Acknowledgements

The authors gratefully acknowledge support from the NRF of Korea Grant funded by the Ministry of Science, ICT and Future Planning (NRF-2013R1A2A1A09014038, 2015M1A2A2074663, 2011-0030254) and Green Science project by POSCO.

## Author contributions

J.H.P. and J.K.K. conceived the idea and designed the experiments. X.S. performed the main experiments and manuscript preparation. H.J, S.J.O. and I.Y.C performed the DBR design and experiments. J.K., I.T.C. and H.K.K. contributed materials/analysis for DSSCs. M.M. and K.Z. analysed the data. J.H.P. and J.K.K. finalized the paper.

## Additional information

**Competing financial interests:** The authors declare no competing financial interests.

**How to cite this article**: Shi, X. *et al.* Unassisted photoelectrochemical water splitting exceeding 7% solar-to-hydrogen conversion efficiency using photon recycling. *Nat. Commun.* 7:11943 doi: 10.1038/ncomms11943 (2016).

