## [Peer review file · Nature Communications]

Reviewers' Comments:

Reviewer #1 (Remarks to the Author)

The authors propose an interesting approach to the basic problem of combining a PEC electrode with a back solar cell: the former should be transparent enough as to allow the solar cell to receive as much light as possible, but at the same time absorb enough light as to efficiently contribute to water splitting. They propose the use of an intermediate optical filter capable of reflecting back blue light to the PEC, hence allowing the use of a more transparent BiVO₄/WO₃ film, while transmitting green and red photons so they can reach the rear dye solar cell. This optical filter should be also conducting to behave as the counter-electrode of the rear solar cell.

The idea is sound, the experimental realization of the materials is seemingly correct and the efficiency reported certainly impressive, but I find the theoretical methods not adequate and the results attained not conclusive. In my opinion, this work is missing a true evaluation of the effect of the 2-layer or 4-layer stack of dense-porous ITO. Would the system not work if only a TiO₂-SiO₂ multilayer is coupled to a single standard dense layer of ITO? Is it not enough the reflection provided by the dielectric multilayer alone? The authors should answer this question before proposing a more complex configuration, like the one reported in the manuscript. Unfortunately, the answer cannot be found in the theoretical evaluation of the optical properties of the structure: they seem to not have considered the extinction coefficient of the materials, but only the real part of the refractive index, which is probably at the origin of the large deviations with respect to the experimental data that is reported in the supporting information. I think this manuscript requires more sound theoretical work and probably some more experiments (system with single ITO layer versus system with layered porous-dense ITO) before publication. If these deficiencies are solved, I would be in favor of publication of this research in Nature Communications.

On other aspects, I have to point out that the authors made a statement that is not correct: "it is nearly impossible to find a combination of two thin-film materials that are conductive and transparent yet have the high refractive index contrast necessary for the desired optical filtering". See for instance "Selectively Transparent and Conducting Photonic Crystals", by GA Ozin, Adv. Mater. 2010, or "Photoconducting Bragg Mirrors based on TiO₂ Nanoparticle Multilayers", by H Miguez, Adv Func Mater 2008 and other related works. Finally, I believe the authors should be more careful with their notation. In Figure 2, S3 and S4, the use of "Absorbance (%)" is incorrect, as absorbance (-logT) is a dimensionless physical magnitude that cannot be expressed as a percentage. Probably they meant "Absorptance".

Reviewer #2 (Remarks to the Author)

This work incorporates a hybrid conductive distributed Bragg reflector (cDBR) to improve the solar-to-hydrogen (STH) conversion efficiency of a tandem PEC cell by photon recycling. The tandem PEC cell was comprised of a FeOOH/NiOOH decorated BiVO₄ photoanode and a Pt decorated DSSC. The reviewer has two major concerns about the work and recommends a major revision before resubmission. First of all, the photocurrents from the photoanodes look suspiciously high for the FeOOH/BiVO₄/WO₃ assembly and the author needs more PEC data to support and explain the high photocurrents. Conventional 3-electrode measurements using real reference electrodes (instead of Pt wire/foil in this study) is needed to quantitatively understand the photoanode performance both under illumination and in the dark. Moreover, spectral response from the photoanode materials is needed to understand the internal quantum yield of the photoanode assembly in order to make the claim that the back-reflecting short-wavelength photons are indeed useful to boost the photocurrent. Also the manuscript does not give a clear description of the two-electrode measurements in Fig. 3. The reviewer suggests a schematic illustration of the electrode connections as an insert in the figure. The placements as well as the size of the Pt wire and Pt foil which were used as the reference/counter electrode were not described, it could have a significant effects on the JV behavior, as a result, the separate JV curves for the individual cells may not reflect the performance of the integrated device. Secondly, the ionic pathways which is necessary to performance un-assisted water-splitting reaction, between the Pt cathode and the photoanode was not described in the manuscript; what is detailed cell configuration including the ionic pathways? What is the resistive loss associated with solution transport and also is there any membrane used to separate the product gases? If not, what is the rate of the product gas crossovers in the described system?

Responses to Reviewer's Comments

Manuscript #: NCOMMS-16-00877-T

Title: Unassisted photoelectrochemical solar water splitting with greater than 7% solar-to-hydrogen conversion efficiency using photon-recycling

Thank you very much for your correspondence and Reviewer's valuable comments on our manuscript (NCOMMS-16-00877-T). We considered the reviewer's comments carefully, and have revised our manuscript accordingly. Please find our response to the Reviewer's comment below:

Reviewer #1 (Remarks to the Author):

(1) The authors propose an interesting approach to the basic problem of combining a PEC electrode with a back solar cell: the former should be transparent enough as to allow the solar cell to receive as much light as possible, but at the same time absorb enough light as to efficiently contribute to water splitting. They propose the use of an intermediate optical filter capable of reflecting back blue light to the PEC, hence allowing the use of a more transparent BiVO₄/WO₃ film, while transmitting green and red photons so they can reach the rear dye solar cell. This optical filter should be also conducting to behave as the counterelectrode of the rear solar cell.

The idea is sound, the experimental realization of the materials is seemingly correct and the efficiency reported certainly impressive, but I find the theoretical methods not adequate and the results attained not conclusive. In my opinion, this work is missing a true evaluation of the effect of the 2-layer or 4-layer stack of dense-porous ITO. Would the system not work if only a TiO₂-SiO₂ multilayer is coupled to a single standard dense layer of ITO? Is it not enough the reflection provided by the dielectric multilayer alone? The authors should answer this question before proposing a more complex configuration, like the one reported in the manuscript. Unfortunately, the answer cannot be found in the theoretical evaluation of the optical properties of the structure: they seem to not have considered the extinction coefficient of the materials, but only the real part of the refractive index, which is probably at the origin of the large deviations with respect to the experimental data that is reported in the supporting information. I think this manuscript requires more sound theoretical work and probably some more experiments (system with single ITO layer versus system with layered porous-dense ITO) before publication. If these deficiencies are solved, I would be in favor of publication of this research in Nature Communications.

Response:

We really appreciate the reviewer's invaluable comments on the theoretical work.

At first, we would like to mention the consideration of extinction coefficients of the materials. As the reviewer pointed out, we found that we had mistakenly calculated the transmittance T without considering the extinction coefficient (k) of the materials, although both the real part (n) and the imaginary part (k) of the complex refractive index of the

materials were used to optimize the cDBR structures by using GA. The transmittance T in the previous manuscript was calculated from the equation,

$$T(\%) = 100(\%) - R'(\%)$$

where the R' is the reflectance calculated without considering the extinction coefficient $k(\lambda)$. We are very sorry about our mistake, and have rectified it as taking measured $k(\lambda)$ into account in the transmittance curves in the revised manuscript (Figure 2) and Supplementary Information (Supplementary Figs S1 and S2).

Figure R1.1 showed the comparison of calculated transmittance with and without considering the $k(\lambda)$ as a function of wavelength. The corrected results show slightly reduced transmittance at the wavelength shorter than ~ 400 nm, but a negligible change at long wavelength range, which is due to the $k(\lambda)$ of ITO layers (Please see Figure R1.2). As the reviewer pointed out, there was a large deviation in transmittance between the calculated and the experimental results even in the wavelength range > 400 nm in Supplementary Fig. 1, which originated from not considering the reflection at the air/glass interface – which will be explained later in this response.

Figure R1.1. Comparison of the corrected simulation results with the previous ones which disregarded the extinction coefficient of the materials for (a) Figure 2a and (b) Figure 2b in the main text.

We note that both the real and the imaginary parts of the complex refractive index of the materials were taken into account during the GA optimization of cDBRs. We believe that it would be better to give a detailed explanation of the GA run by MATLAB. The transfer matrix method was used for calculating the reflectance R and the transmittance T of each cDBR. Transfer matrix of each layer is:

$$M(j) = \begin{bmatrix} m_{11} & m_{12} \\ m_{21} & m_{22} \end{bmatrix} = \begin{bmatrix} \cos\beta_j & -\frac{i}{p_j} \sin\beta_j \\ -ip_j \sin\beta_j & \cos\beta_j \end{bmatrix}$$

where $\beta_j = \frac{2\pi}{\lambda_0} n_j h_j \cos\theta_j$ (by refraction, $n_j \sin\theta_j = n_{j-1} \sin\theta_{j-1}$), $p_j = n_j \cos\theta_j$ (for TE – polarized light), $p_j = \frac{n_j}{\cos\theta_j}$ (for TM – polarized light), n_j , h_j , θ_j are complex refractive index, height, and incident angle of j^{th} layer, respectively, and λ_0 is the wavelength of incident light [*Principles of Optics, M. Born and E. Wolf, 7th ed. (Cambridge Univ. Press, New York, 1999)*]. The complex refractive index including index of refraction (\mathbf{n}) and extinction coefficient (\mathbf{k}) was used to take the absorption of light by each layer into account. Then, the calculated reflection and transmission coefficients were:

$$r = \frac{(m_{11} + m_{12}p_l)p_1 - (m_{21} + m_{22}p_l)}{(m_{11} + m_{12}p_l)p_1 + (m_{21} + m_{22}p_l)}, \quad R = |r|^2$$

$$t = \frac{2p_1}{(m_{11} + m_{12}p_l)p_1 + (m_{21} + m_{22}p_l)}, \quad T = \frac{p_l}{p_1} |t|^2$$

where ‘1’ and ‘ l ’ denote the first and last layer of multilayer films. GA optimization calculations begin with the generation of a population of multilayer cDBR films with a fixed number of layers whose thicknesses and compositions are randomly generated, followed by the evaluation of the FOM of each member. The detailed explanation of the GA optimization calculation as described above has been added in the Supplementary Information.

In order to minimize the deviation between calculated and experimentally measured transmittance (and reflectance) of cDBRs, we used measured optical parameters of the films we fabricated. In detail, the refractive index (\mathbf{n}) and the extinction coefficient (\mathbf{k}) of dense and porous ITO, SiO₂, and TiO₂ were measured by ellipsometry. Two parameters – amplitude ratio ($\tan\Psi$) and phase difference (Δ) – were collected and fitted by a simulation using quantum mechanically derived dispersion to get an information of the film such as \mathbf{n} , \mathbf{k} , and film thickness (\mathbf{h}) as shown in Figure R1.2a and b [*Handbook of Optical Constants of Solids II, E. D. Palik, ed. (Academic Press, Toronto, 1991)*]. As a result, \mathbf{n} and \mathbf{k} were obtained and used for our calculations. Figure R1.2c shows \mathbf{n} and \mathbf{k} curves of dense and porous ITO as a function of wavelength. For the wavelength $> \sim 400$ nm, the \mathbf{k} curves of both dense and porous ITO are almost zero, indicating a negligible absorption by the layers, where as in the wavelength range $< \sim 400$ nm, the \mathbf{k} values increases with decreasing wavelength, resulting in a slightly reduced transmittance in comparison with the results without considering \mathbf{k} , as shown in Figure R1.1. To clarify the simulation work, we have added the following statements, “The measured refractive indices and the extinction coefficients of the dense and the porous ITO layers deposited by OAD were used for the optimization of the cDBRs (Supplementary Fig. 1a).” in the Methods section of the revised manuscript, and “For GA optimization, the measured refractive index as well as the extinction coefficient profiles of the dense ($\theta_{\text{OAD}} = 0^\circ$) and porous ($\theta_{\text{OAD}} = 70^\circ$) ITO layers fabricated by OAD were used (Supplementary Fig. 1a).” in Supplementary Information. In addition, we have revised the Supplementary Fig.1a to include both the refractive index (solid lines) and the extinction coefficient (dashed lines) of dense ITO and porous ITO as a function of wavelength in Supplementary Information.

Figure R1.2. The phase difference (Δ) and amplitude ratio ($\tan\Psi$) measured by ellipsometry at 70.2° incident angle (dotted lines) and model fittings (solid lines) for (a) dense ITO ($\theta_{\text{OAD}}=0^\circ$) with standard deviation of 3.46 and (b) porous ITO ($\theta_{\text{OAD}}=70^\circ$) with standard deviation of 2.89. (c) The refractive index (solid lines) and the extinction coefficient (dashed lines) of dense ITO and porous ITO as a function of wavelength

Secondly, let us try to explain the origin of a large deviation in transmittance between the calculated and the experimental results in Supplementary Fig.1 even in the wavelength range $> \sim 400$ nm where the absorption by the materials is negligible. In the GA optimization simulation, it was assumed that the incident light propagates from the glass substrate with infinite thickness, as shown in Figure R1.3a. However, in the real system with which the experimental transmittance values were measured, the incident light propagating from the air, not from the inside of infinitely thick glass substrate, meets the air/glass substrate interface where reflection and transmission occur, and transmitted light passes through the glass substrate with finite thickness, as shown in Figure R1.3b. We should have considered the reflection of the incident light at the air/glass substrate interface so as to precisely evaluate the experimental results. Since the reflection was not considered in the previous calculation, the results showed a large deviation in transmittance even in the long wavelength region in Supplementary Fig.1, which have been rectified (Supplementary Fig.1b and d, and Supplementary Fig.2b) in the revised manuscript. Figures R1.4 showed the comparison between previous results and corrected results considering $k(\lambda)$ as well as the reflection at the air/glass substrate. Reduced transmittance T by considering the reflection as well as the absorption results in reduced deviation in T at both long (>400 nm) and short (< 400 nm) wavelength ranges, respectively. However, the deviation between the experimental curve and the corrected calculation result for the 7-layer ITO cDBR shown in Figure R1.4b is still not negligible, which is due to the imperfect interface between individual porous and dense ITO

layers, as described in the manuscript. Likewise, the calculated transmittance of the dielectric DBR stacks becomes slightly reduced compared to the previous one as considering the reflection (Figure R1.4c). We have corrected the graphs in the revised manuscript. Again, we really appreciate the reviewer's critical and invaluable comments.

Figure R1.3. (a) The system used for the GA simulation, in which the light is assumed to be incident from the glass substrate with infinite thickness. (b) Schematic description of the real system with which the transmittance is experimentally measured: the light is incident from the air and partially reflected at the air-glass interface.

Figure R1.4. Comparison of the corrected simulation results with the previous ones which disregarded the extinction coefficient of the materials as well as the reflection loss at the glass surface for (a) Supplementary Fig.1b, (b) Supplementary Fig.1d and (c) Supplementary Fig.2b in Supplementary Information.

Next, let us try to answer the reviewer's questions: *Would the system not work if only a $\text{TiO}_2\text{-SiO}_2$ multilayer is coupled to a single standard dense layer of ITO? Is it not enough the reflection provided by the dielectric multilayer alone?* The questions are very critical and important, and there should have been an appropriate explanation on this issue in the manuscript. The cDBRs in this study have two main functions; an optical filter as well as a counter-electrode of the rear DSSC. In order to be used as a counter-electrode, the cDBR should be conductive, which is the reason why a dielectric-multilayer-only filter has not been considered. As the reviewer pointed out, a cDBR consisting of a $\text{TiO}_2\text{-SiO}_2$ multilayer

coupled to a single standard dense ITO layer would fulfil the two functions, however, it has not been experimentally tried due to the following reasons. At first, the cDBR with single dense ITO layer was also optimized by using GA, and compared with the cDBRs with 2-layer and 4-layer ITO. Figure R1.5a showed the simulated transmittance data of three cDBRs. The cDBR with 2-layer and 4-layer ITO showed higher FOM of 1.3186 and 1.3238, as described in the main text, than the optimized structure with single dense ITO layer (FOM = 1.3037), meaning that the optical performance of the cDBRs with 2- and 4-layer ITO stacks (with porous ITO on the top) was superior to that of the cDBR with optimized single dense ITO layer.

Figure R1.5. (a) Simulated transmittance of cDBRs with single dense ITO, 2-layer ITO, and 4-layer ITO. (b) The reduction curve of cyclic voltammetry (CV) measurement for Pt coated cDBR with single dense ITO and porous/dense ITO. Pt deposition condition was following the process demonstrated in the experimental DSSC fabrication part with 20 times diluted electrolyte used for measurement.

In addition to the optical benefit, a porous layer was intentionally chosen as the topmost layer in this study because we believe that a porous framework for Pt coating is beneficial for a high performance of DSSC enabled by a higher surface area and electro-catalytic activity than the planar one (*Nanoscale*, 2013, 5, 4951–4957). In order to experimentally prove the enhanced electrocatalytic effect of Pt on the ITO structures with top-porous layer, the cyclic voltammetry (CV) measurements of the single dense ITO layer and the 2-layer porous/dense ITO deposited on the glass substrate were performed. As shown in Figure R1.5b, under the same condition, the 2-layer ITO shows higher cathodic peak current density, which indicates that the cDBR with top-porous ITO layer has better electrocatalytic ability for electrolyte reduction. In response to the reviewer’s valuable comments and to clarify the strengths of layered ITO stacks, we have added the following statements and additional Supplementary Fig. 3 (same as Figure R1.5): “Supplementary Fig.3a show the calculated transmittance of the hybrid cDBR structures with single dense ITO top layer, and 2-layer, and 4-layer porous/dense ITO stacks. The cDBR with 2-layer and 4-layer ITO showed higher FOM of 1.3186 and 1.3238, respectively, than that with single dense ITO layer (FOM = 1.3037), indicating that the optical performance of the cDBRs with 2- and 4-layer ITO stacks (with porous ITO on the top) is superior to that of the cDBR with optimized single dense ITO layer. In addition to the optical benefit, a porous layer was intentionally chosen as the topmost layer for experiments because a porous framework for Pt coating would be beneficial for a high performance of DSSC enabled by a higher surface area and electrocatalytic activity than the

planar one⁴ (Supplementary Fig. 3b). As a result, the hybrid cDBRs with 2-layer and 4-layer ITO stacks were fabricated as described in the main text.

[4] Bao, C. *et al.* The maximum limiting performance improved counter electrode based on a porous fluorine doped tin oxide conductive framework for dye-sensitized solar cells. *Nanoscale* **5**, 4951–4957 (2013) ”

(2) On other aspects, I have to point out that the authors made a statement that is not correct: "it is nearly impossible to find a combination of two thin-film materials that are conductive and transparent yet have the high refractive index contrast necessary for the desired optical filtering". See for instance "Selectively Transparent and Conducting Photonic Crystals", by GA Ozin, Adv. Mater. 2010, or "Photoconducting Bragg Mirrors based on TiO2 Nanoparticle Multilayers", by H Miguez, Adv Func Mater 2008 and other related works.

Finally, I believe the authors should be more careful with their notation. In Figure 2, S3 and S4, the use of "Absorbance (%)" is incorrect, as absorbance (-logT) is a dimensionless physical magnitude that cannot be expressed as a percentage. Probably they meant "Absorptance".

Response: Thank you very much for the comments. In accordance with the reviewer’s comments, we have revised the statement as “A number of studies on the combination of two thin-film materials that are conductive and transparent, yet have a high refractive-index contrast have been performed, however, their application in PEC tandem cell with desired optical and electrical properties is still immature and needed to be further studied”. And the literatures mentioned by the reviewer have been added in the reference. In addition, all “Absorbance (%)” has been corrected to “Absorptance (%)”.

Reviewer #2 (Remarks to the Author):

(1) This work incorporates a hybrid conductive distributed Bragg reflector (cDBR) to improve the solar-to-hydrogen (STH) conversion efficiency of a tandem PEC cell by photon recycling. The tandem PEC cell was comprised of a FeOOH/NiOOH decorated BiVO4 photoanode and a Pt decorated DSSC. The reviewer has two major concerns about the work and recommends a major revision before resubmission. First of all, the photocurrents from the photoanodes look suspiciously high for the FeOOH/BiVO4/WO3 assembly and the author needs more PEC data to support and explain the high photocurrents. Conventional 3-electrode measurements using real reference electrodes (instead of Pt wire/foil in this study) is needed to quantitatively understand the photoanode performance both under illumination and in the dark. Moreover, spectral response from the photoanode materials is needed to understand the internal quantum yield of the photoanode assembly in order to make the claim that the back-reflecting short-wavelength photons are indeed useful to boost the photocurrent.

Response: Thanks for the deep concern of PEC performance part. As the reviewer's comment, this is quite important information, so we showed more detailed PEC measurement information under 3-electrode system below, by using the arithmetic mean value from the curves of 5 samples for each condition. The error bar illustrated the range of highest and lowest value among the 5 samples at around 1.23V.

Figure R2.1. The PEC performance of the photoanodes under 3-electrode system

And also we summarized the average photocurrent value of key point potential for different condition samples under different measurements

Measurement condition	3-electrode system, at 1.23V vs. RHE		2-electrode system at operating potential U_{op}	
	WO ₃ /doped BiVO ₄	WO ₃ /doped BiVO ₄ /FeOOH/NiOOH	WO ₃ /doped BiVO ₄ /FeOOH/NiOOH	WO ₃ /doped BiVO ₄ /FeOOH/NiOOH with DBR stack
Current Density (mA/cm ²)	2.9	4.3	4.6	5.7

From the results it can be seen the WO₃/doped BiVO₄ has a current density 2.9 mA/cm² at 1.23V vs. RHE. This is common value for reported WO₃/BiVO₄ *n-n* heterojunction with W, Mo doping, which could be attributed to the efficient charge separation and transport process induced by the compact embedded structure used for *n-n* heterojunction (*Nano Energy* 2015, 13, 182–191), as well as the proper element doping (*Energy Environ. Sci.*, 2016, *Advance Article*; DOI: 10.1039/C6EE00129G). Then, the optimal amount of FeOOH/NiOOH cocatalyst was used (further enhance the surface charge transfer property) and the current density is around 4.3 mA/cm² at 1.23V vs. RHE. Then the samples were checked under 2-electrode system and the current density was a little higher than the ones checked under 3-electrode system, which owned 4.6 mA/cm² at the operating potential U_{op} , around 0.9V vs. Pt in this work. Similar results between 2-electrode and 3-electrode have also been mentioned

and explained by previous report (*Nat. Commun.*, **4**, 2195, (2013), *supplementary note 3*). After the use of cDBR (elevate the light harvesting) this value got further enhanced which is the main focus of this work.

For the spectral response from the photoanode materials with/without DBR, we have given the related discussion in supplementary material II as well as in Supplementary Figs 4 a-c. But still, we think the comments of reviewer about the quantum yield is quite reasonable and necessary in this work to make the work more perfect. Therefore, the EQE data under 1.23V vs. RHE have been checked (we think here the reviewer actually means the “external quantum yield”, rather than “internal quantum yield”, since in this work we mainly investigate the light cycling effect induced from the addition of cDBR, but the internal quantum yield is obtained from the external quantum yield dividing by the light absorption, which expresses how good for the device to use the absorbed photons, and it has excluded the light cycling effect for the light harvesting itself, therefore it is out of scope of this work), and the result is shown below.

Figure R2.2. The external quantum efficiency of the photoanode with/without cDBR

The EQE data showed that the addition of cDBR could act a positive role for the enhancement of external quantum yield, in the meanwhile this result basically agrees well with the analyses in the Supplementary Information II. Since this is also one of the important supporting data for this work, based on reviewer’s suggestion we have added it in the supplementary information as Supplementary Fig. 4d.

(2) Also the manuscript does not give a clear description of the two-electrode measurements in Fig. 3. The reviewer suggests a schematic illustration of the electrode connections as an insert in the figure. The placements as well as the size of the Pt wire and Pt foil which were used as the reference/counter electrode were not described, it could have a significant effects

on the JV behavior, as a result, the separate JV curves for the individual cells may not reflect the performance of the integrated device. Secondly, the ionic pathways which is necessary to performance un-assisted water-splitting reaction, between the Pt cathode and the photoanode was not described in the manuscript; what is detailed cell configuration including the ionic pathways?

Response: In below figure, we introduced our two-electrode measurements setup for Figure 3 by using the following brief schematic diagrams (all in section views).

To obtain J-V curves of front PEC photoanodes with and without DBR layer, we used a conventional measurement method (Figure R2.3), in which the J-V curves of front PEC photoanodes could not be influenced by back DSSCs. Under 1sun illumination, photoanode (with/without cDBR on the back) which was exposed to the water electrolyte with confined area was connected with the lead of “working electrode” from the potentiostat, and the Pt foil was used as both “reference electrode” and “counter electrode”.

Figure R2.3. the configuration of photoanode two-electrode measurement

To obtain J-V curves of the rear DSSC as shown in Fig. 3, the tandem device was prepared as shown in Figure R2.4. If we connect the dye-sensitized TiO_2 electrode and the Pt/DBR counter electrode with “working electrode” and “reference/counter electrode” of potentiostat, respectively J-V curves of the rear DSSC could be obtained. Even though the tandem cells were under 1sun illumination, the rear DSSCs can absorb only the filtered solar light by the front PEC cells.

Figure R2.4. the configuration of DSSC two-electrode measurement

To check J-V curves of the tandem devices, we connected the photoanode and Pt/DBR by conductive silver paste, which induce electron transfer from the PEC photoanode to Pt/DBR. Before immersing into the electrolyte, all the silver paste was covered by super glue and dried. To connect the tandem device with potentiostat, the dye-sensitized TiO₂ electrode was connected with “working electrode” of potentiostat and Pt foil were connected with “reference/counter electrode” of potentiostat (Figure R2.5). The dimension of the used Pt foil was 1.5cm × 1.5cm. We added this information in our revised manuscript “Methods” section.

Figure R2.5. the configuration of tandem device two-electrode measurement

Additionally, the detailed internal configurations as well as the charge carrier pathways for our tandem device is demonstrated in Figure R2.6 (which applies to both devices with and without the addition of DBR). As it is marked in the third schematic above, when we did 2-electrode measurement for the whole tandem device, the working electrode lead is connected to the anode part of DSSC (Dye/TiO₂). Once light is incident to the photoanode, photoexcited electrons from the photoanode will be transferred through the silver paste to the back side (Pt/DBR counter of DSSC). The remained holes oxidise OH⁻ to oxygen, which means the O₂ generation occur at our photoanode part. The transferred electrons to Pt counter are used to reduce (Co(bpy)₃)^{2+/3+} for the DSSC electrolyte. In the meanwhile, long wavelength light which was not absorbed by the photoanode could penetrate the front cell and reach the dye/TiO₂ of the DSSC, where the electrons in the dye molecules are excited from the HOMO (S⁰) to the LUMO (S^{*}), and those excited electrons will be transferred through the external circuit to the Pt foil (work as the counter electrode for the tandem device measurement), and used for hydrogen generation. The remained holes in dyes could be accepted by (Co(bpy)₃)^{2+/3+}.

Figure R2.6. charge carrier pathways for our tandem device

(3) What is the resistive loss associated with solution transport and also is there any membrane used to separate the product gases? If not, what is the rate of the product gas crossovers in the described system?

Response: This is really an important issues for the commercialization aspect. In this work, we didn't use membrane for separating the product gases. As mentioned in your comment, the membrane-free cell is not beneficial for efficient and safe gas separation/collection.

For the estimation of solution loss in our case, the value of the membrane-free cell with the buffered to near-neutral pH could be estimated to be pretty small (less than 25mV at 10mA/cm², which could be even smaller at ~5mA/cm² for our case), which is based on the previous report (*Energy Environ. Sci.*, 2015, 8, 2760). In addition, our system represented about 96.8% and 97.1% Faraday efficiency for the tandem devices with and without cDBR at 120mins, which means our system don't have serious resistive loss associated with solution transport.

During gas production test as shown in Fig. 3(c), we did not separate the evolved H₂ and O₂ from the reactor in order to get more accurate overall evolved amount. So, we could not know the rate of the product gas crossovers. Based on reviewer's comments, we have characterized the rate of the product gas crossovers via using T-structured cell, from which we could observe the rate of the product gas crossovers (Figure R2.7). At the time of 120 mins, around 11.6 mol% O₂ in cathode chamber (Pt) was detected, while there's 35.7 mol% H₂ in anode chamber (photoanode part). Undeniably, the gas crossovers are not low enough in this work (*Energy Environ. Sci.*, 2014, 7, 3371–3380). In addition, from that the O₂ value in cathode chamber and H₂ value in anode chamber keep increasing during 120mins it can be deduced that the crossovers keep occurring during the whole measurement process.

In this study, the main focus of our work is the investigation of the DBR function to tandem device, in which the gas evolution acts as an effective way for characterizing the performance of the tandem devices with/without DBR. Hence, we didn't devote that much of our effort to the optimal gas separation work itself. Nevertheless, preventing product-gas crossovers is

quite important in view of large scale production. We'll try to make some progress on this part in our following work.

Figure R2.7. the gas evolution individual measurement from two chambers for the tandem device with 2-layer cDBR.

Since the references mentioned in the last part (part (3)) are extremely useful to improve our work quality, we have cited them in the revised manuscript.

Reviewers' Comments:

Reviewer #1 (Remarks to the Author)

I am satisfied with the authors' thorough and detailed response to my comments. This manuscript is now ready for publication in Nature Communications.

Reviewer #2 (Remarks to the Author)

The author has made significant improvements in the revision and the reviewer recommend publication of the work.